# Unravelling the Role of PARP1 in Homeostasis and Tumorigenesis: Implications for Anti-Cancer Therapies and Overcoming Resistance

**DOI:** 10.3390/cells12141904

**Published:** 2023-07-21

**Authors:** Taylor Lovsund, Fatemeh Mashayekhi, Amira Fitieh, James Stafford, Ismail Hassan Ismail

**Affiliations:** 1Division of Experimental Oncology, Department of Oncology, Faculty of Medicine & Dentistry, University of Alberta, 11560 University Avenue, Edmonton, AB T6G 1Z2, Canada; tklovsun@ualberta.ca (T.L.); fmashaye@ualberta.ca (F.M.); 2Department of Biophysics, Faculty of Science, Cairo University, Giza 12613, Egypt; amohamed@sci.cu.edu.eg; 3Department of Biological Sciences, Faculty of Science, University of Alberta, Edmonton, AB T6G 2E1, Canada; stafford@ualberta.ca

**Keywords:** genome instability, DNA repair, PARP1, PARPi resistance, PARylation, homologous recombination

## Abstract

Detailing the connection between homeostatic functions of enzymatic families and eventual progression into tumorigenesis is crucial to our understanding of anti-cancer therapies. One key enzyme group involved in this process is the Poly (ADP-ribose) polymerase (PARP) family, responsible for an expansive number of cellular functions, featuring members well established as regulators of DNA repair, genomic stability and beyond. Several PARP inhibitors (PARPi) have been approved for clinical use in a range of cancers, with many more still in trials. Unfortunately, the occurrence of resistance to PARPi therapy is growing in prevalence and requires the introduction of novel counter-resistance mechanisms to maintain efficacy. In this review, we summarize the updated understanding of the vast homeostatic functions the PARP family mediates and pin the importance of PARPi therapies as anti-cancer agents while discussing resistance mechanisms and current up-and-coming counter-strategies for countering such resistance.

## 1. Introduction

The human genome strives to maintain integrity in the face of perpetual genotoxic stress enduring both exogenous agents and endogenous factors which threaten the survival of the cell [1]. To defend against this constant barrage, cells have devised a number of response mechanisms involved in the detection, signaling and resolution of stress. In the event of irreparable damage or an insufficient response, programmed cell death is initiated. Long established as key players in this process is the PARP family, but the extent of their roles across the cell is still coming to light.

The current PARP family consists of 17 enzymes, PARP1 through PARP16 (Table 1) [1,2,3,4,5,6,7,8], homologous for a catalytic domain containing ADP-ribosyl transferase activity via an NAD+ substrate [9,10]. In addition to their catalytic domain, each PARP member has distinct regions related to their specific cellular functions [3]. The reversible post-translational transfer of ADP-ribose, known as ADP ribosylation (ADPr), targets a variety of proteins, including PARP itself, as well as nucleic acids to fulfil its multitudinous role across biological tasks [9,10]. The catalysis of ADPr can be divided into the synthesis of polymers of ADP-ribose known as PAR or mono-ADP ribose known as MAR. Table 1 identifies this critical distinction necessary for examining both the structural and biological functions of each PARP family member [9,11].

While PARP1, 2, 5a and 5b transfer PAR to their target molecules, PARPs 3-16 transfer MAR, thus distinguishing the proteins as PAR or MAR transferases (PARTs and MARTs, respectively), with the exception of PARP 9 and 13, which appear to be catalytically inactive, as seen in Table 1 [9]. The catalytic domain of all PARP family members includes both the ADP-ribosyl transferase subdomain as well as a helical subdomain involved in the autoinhibition of NAD+ binding [15,16]. The PARylating subset of the PARP family possesses HYE motifs, responsible for the lengthening of PAR chains on target molecules [17]. PARylators have shown their involvement in the regulation of cell division, apoptosis, DNA damage detection and resolution, while further functions have yet to be uncovered [3,17,18].

Although most PARP enzymes are MARTs, the characterization of their functions was elusive for years, largely due to the difficulty of selectively inhibiting them with small molecules [3,18]. Fortunately, advancements in medicinal chemistry have allowed for the production of these inhibitors, for which PARP10 has been shown to be a promising clinical candidate [19]. All MARylating PARPs feature an HYI, HYL or HYY, except PARPs 3 and 4, which, interestingly, only produce MAR despite containing HYE domains [17,20]. MARTs also commonly share WWE motifs, CCCH zinc fingers and RNA recognition motifs linking to their regulatory roles in RNA metabolism, the actin cytoskeleton and the cell cycle [17].

## 2. The Regulation of Cellular Homeostasis by PARP Enzymes

As previously mentioned, the homeostasis of the cell is largely dependent on the activities of the PARP family [9]. PARP 5a and 5b, commonly known as tankyrase 1 and 2, have been continually shown to regulate telomere length and spindle assembly during the metaphase, which is vital for maintaining chromosomal stability among proliferating cells [20,21]. PARP7 has implications in the cell cycle, with a high-affinity RNA binding domain indicating a possible role in transcription [9]. PARP7 was also noted for its regulatory role in both innate immunity and transcription factor function when protein depletion led to a decreased rate of mitosis [9,22]. PARP14 is largely thought to regulate cytoskeleton formation and motility, while PARP16 has been implicated in endoplasmic reticulum stress responses [9]. These roles are just a subset of PARP’s vast responsibility across the cell, and new contributions continue to come to light. This light has shed a particular brightness on the responsibilities PARP has in sensing and regulating the repair of damaged DNA in the pursuit of genomic stability.

## 3. PARP-Mediated Regulation of Single-Strand Break (SSB) Repair

It has been well established that PARP1, 2 and 3 are key players in the cellular response to DNA damage, sharing a WGR domain that regulates the protein’s response through a direct interaction with the damaged DNA [1,3]. PARP2 and PARP3 favor activation by a 5’ phosphate group of damaged DNA, while PARP1, which has additional Zn1, Zn2, Zn3 zinc finger and BRCT domains, has no preference [3]. Recent studies have also indicated PARP3’s ability to directly MARylate 5’ and 3’ terminal phosphate residues of DNA [23,24]. In addition to its role in sensing DNA damage, the PARylation catalyzed by PARP1 is crucial in the recruitment and activity of repair factors and chromatin remodelers working to preserve genomic stability [1,3].

SSB can occur due to a number of exogenous and endogenous factors resulting in PARP1 recognizing the SSB and binding to the damaged DNA via its Zn1, Zn3 and WGR domains (Figure 1) [1]. As seen in Figure 1, after the initial binding, PARP1 undergoes several changes, including the binding of its Zn2 domain to form a complex with the DNA and the subsequent unfolding of its HD domain, resulting in enzymatic activation [1]. Once bound and activated, PAR polymer synthesis begins the enzyme’s vast role in SSBs, including the self-PARylation and BRCT domain-assisted recruitment of scaffold protein XRCC1 required for the resolution of DNA breakage via the common base excision repair (BER) pathway [1,10,25].

BER is required to repair DNA that has been subject to base removal and UV damage, among other factors, all of which create an apurinic/apyrimidinic (AP) site [26]. Both PARP1 and PARP2 have been implicated in the recognition of these AP sites, which is followed by DNA cleavage by APE1. However, it has been suggested that PARP1/2 are also able to perform strand incision by their 5′ deoxyribose-5-phosphate/AP (5′dRP/AP) [27]. After cleavage, PARP enzymes dissociate from the DNA through an auto modification, leading to the recruitment and activation of the BER complex on AP sites, as seen in Figure 1 [26]. The BER complex is then able to complete the repair by excising the base with a DNA glycosylase, followed by the insertion and ligation of the correct base with DNA Polymerase β and ligase 3, respectively [28].

After UV-induced DNA damage, detection and repair can also occur via the nucleotide excision repair (NER) pathway, in which PARP1 has been shown to interact with key members [29]. A key scaffold protein, Xeroderma Pigmentosum Complementation Group A (XPA), associates with PARP1 after UV exposure, promoting XPA binding to chromatin. PARP1 has also been shown to associate with and regulate DDB2, a major component of the NER pathway, affecting DDB2’s affinity at DNA lesions and its ability to later recruit chromatin remodeler ALC1 [30]. NER-driven lesion removal, DNA gap filling and repair are then completed through coordination between proteins in the pathway including XPA [31].

**Figure 1 cells-12-01904-f001:**
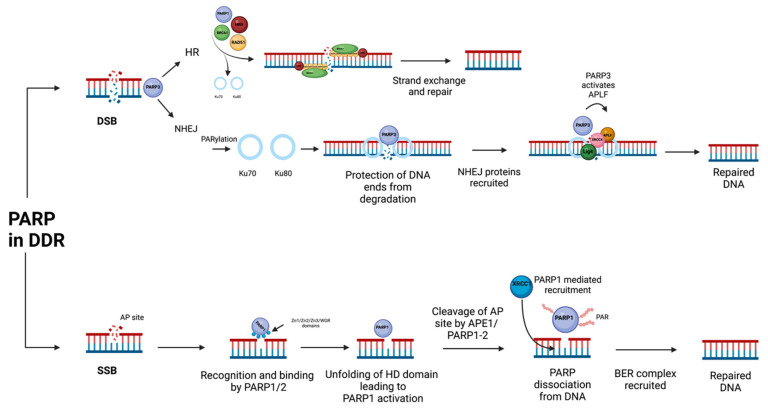
PARP helps mediate the choice with SSB and DSB repair pathways. PARP1 is required for the detection of DSBs and recruits MRE11 (part of the MRN complex), BRCA1 and RAD52 [32]. PARP1 competes with Ku for bind sites on DSBs, shifting the outcome towards HR [33]. PARP3 has been shown to promote NHEJ through interactions with Ku70 and Ku80, which then interact with DNA-dependent protein kinase catalytic subunit (DNA-Pkcs) to initiate the repair of the DSB [34,35]. In SSB repair, PARP1/2 recognizes AP sites and interacts with the DNA, leading to enzymatic activation and subsequent AP site cleavage [26,27]. PARP1 then dissociates from the DNA and recruits BER proteins responsible for damage repair.

## 4. PARP-Mediated Regulation in DDR

The DNA damage sensing ability of PARP1 extends past the gaps of SSBs, capable of detecting the overhanging and blunt ends of double-strand breaks (DSBs) [36]. Cells respond to DSBs by activating one of two major repair pathways, Non-Homologous End Joining (NHEJ) or Homologous Recombination (HR), both of which are regulated by members of the PARP family, with PARP3 playing a decisive role in the pathway choice (Figure 1) [1,23]. The initial recognition of PARP1 in SSBs is like that of DSBs [36]. The zinc finger, WGR and catalytically active domains of PARP1 form a complex with DSBs to activate and initiate its array of roles within the DNA damage response.

NHEJ, the quicker, more frequent and error-prone DSB repair mechanism, is promoted by the MARylator, PARP3, which interacts with proteins in the NHEJ pathway and accelerates the repair [1]. PARP3 is required in multiple phases of the response, interacting with Ku70-Ku80 to prevent nuclease degradation in an early stage, thus encouraging the NHEJ pathway while also improving efficacy [34]. The C terminal region of Ku80 helps to recruit and retain DNA-PKcs at DSBs to initiate repair [35]. In later stages, the PAR-dependent interaction with APLF mediates the phosphorylation of APLF on Ser 116 by ATM, which helps recruit the protein to damaged DNA, where APLF can promote XRCC4/DNA ligase IV-mediated litigation [1,37]. These roles drive the cell’s decision to follow the NHEJ pathway, rather than HR.

In the event the HR pathway is activated, PARP1 channels cells to HR by interfering with proteins necessary for NHEJ, such as the Ku heterodimer and Ligase IV (Figure 1) [33]. After Ku removal, PARP1 mediates the recruitment of the MRN (Mre11/Rad50/Nbs1) complex to process the ends of the DNA, BRCA1 to control DNA resection and RAD51 loading to mediate strand exchange [32,38]. Specifically, BRCA1 becomes ADP-ribosylated by PARP1, restricting end resection at the DSB by hindering BRCA2 and EXO1 recruitment through a loss of affinity of BRCA1 to the DNA [39]. The recruitment of key damage mediators such as MDC1 and RAD51 is then dependent on PARP5a and PARP5b (tankyrases) through direct binding and end resection dictation, respectively [40]. MDC1 mediates the recruitment and retention of RAD51 on the chromatin through a direct interaction with RAD51 [41]. The crucial formation of the BRCA1A complex has been shown to rely on the PARylation of PTEN, leading to the activation of the AKT pathway [1]. BRCA1 loading onto damaged DNA occurs via tankyrase promotion and stabilization [40]. While the ADP-ribosyltransferase domain is not responsible for this particular function, it remains an intrinsic property of the protein. The efficient repair of double-strand breaks (DSBs) necessitates the accessibility of recruited repair factors to damaged DNA upon the activation of homologous recombination (HR). Consequently, comprehending the process of PARP1-catalyzed chromatin reorganization is crucial [1].

## 5. PARP-Mediated Chromatin Reorganization

In response to DNA lesions, the activation of PARP1 leads to the PARylation of serine residues on core histones and facilitates early-phase histone removal at the site of DNA damage [42,43]. PARylation, along with the negative charge of the polymeric chain, signals for chromatin relaxation, allowing for the recruitment and processing of chromatin remodelers and DNA damage repair proteins [42]. Also helping to facilitate chromatin reorganization is a histone demethylase, Kdm4b, which is important for the phosphorylation of ATM substrates and dependent on PARP1 for recruitment to the damage site [44,45]. Kdm4b has been implicated in conferring a survival advantage due to its role in the DNA damage response. Upon its depletion, disturbances in the functions of the DDR proteins RAD51 and P53 were found, thus compromising the integrity of the DSB repair. Additionally, the ATP-dependent chromatin remodeling enzyme (ALC1) has been shown to recognize PARylation at sites of DNA damage [43]. The later recruitment of DNA damage factors through the ubiquitylation of histone H2AX and an E3 ubiquitin ligase RNF168 is also mediated through PARP1 [42,43]. Through its involvement in chromatin relaxation and condensation alongside its DDR response, PARP1 hints at the vast extent of its role in allowing cells to shift from replication to repair and back again [1]. 

## 6. PARP’s Role in the Stalling and Protection of Stressed Replication Forks

Although DNA repair has been in focus, new insights into the role of PARP in replication fork stability and protection are becoming increasingly important in understanding the cellular mechanisms that maintain genomic integrity [46]. Replication stress can be described as interference within the competence of the cell cycle to faithfully regulate chromosome division, leading to the slowing or stalling of the replication fork [47]. This can arise from numerous exogenous and endogenous factors often compromising cell cycle components such as fidelity and speed. DNA damage repair in cells undergoing replication stress requires the stalling and stabilization of active replication forks (RF) while subsequently protecting the nascent DNA from uncontrolled degradation to prevent genomic instability.

Recent publications have shown that PARP1 and PARP2 are required to repair DNA DSBs arising from the collision of RFs with unrepaired SSBs [46]. Upon replication stress-induced DSBs, PARP1 and PARP2 contribute to the loading and stabilization of Rad51, a crucial intermediate of HR, by antagonizing the anti-recombinogenic activity of Fbh1 [46].

Another target of PARP1 recruitment is the RECQ1 helicase, which is critical for the protection of stalled RFs [48]. RECQ1 associates with proliferating cell nuclear antigen (PCNA) and PARP1 to recruit XRCC1 to the site of damage, thus promoting fork repair and stability. Once the damage is resolved, the stalled RF must be restarted for the cell to progress into the G2/M phases of the cell cycle [49]. To do so, PARP1 binds to stalled RFs and recruits MRE11, a nuclease involved in end resection that is required for replication restart [49].

PARP10 binding to ubiquitinated PCNA has been shown to be a requirement for restarting stalled forks via translesion DNA synthesis (TLS), which allows synthesis machinery to bypass the fork structures by providing low-fidelity polymerases carrying modified DNA bases [24]. Although PCNA is responsible for restarting forks undergoing DNA damage, the TLS method of restart creates more error-prone DNA replication and can result in increased genomic instability [24,50]. Despite this, evidence of direct PARP10 activity at RFs is lacking, most likely due to a struggle in discerning MAR in experiments [51].

Similar to PARP10, PARP14 has been shown to reduce replication stress through the promotion of HR-mediated repair via PCNA association during the S phase of the cell cycle [52]. In the absence of PARP14, studies have shown deficient HR-mediated repair, defined by persistent RAD51 foci [52]. MARylation by PARP14 is shown to help in RAD51 removal, allowing for the completion of HR [53]. Importantly, PARP14 was also found to restrain common fragile sites (CFS), which are prone to strand breakage, threatening genomic stability and leading to RF stalling [52]. Through its interactions with CFS and its role in HR, PARP14 is suggested to play a critical part in maintaining genomic stability during normal and stressed replication environments.

The determination of PARP’s role in the absence of exogenous stress has challenged researchers for years. Recently, an increase in PARP-1 mediated- PARylation events has been detected during the S phase of the cell cycle due to the presence of unligated Okazaki fragments [54]. In each S phase, about 30–50 million Okazaki fragments are formed, and their ligation is essential in maintaining genomic integrity [51,55]. It has been indicated that the presence of PARP1 is required for the detection and signaling of lagging strand fragments that have evaded the standard ligation pathway [54,55]. Strand fragments that have evaded ligation are then repaired by SSB proteins such as XRCC1 and DNA Ligase 3, which require PARP1 for recruitment. Altogether, PARP enzymes have been heavily studied in the replication stress response, confirming their crucial role in leading to genomic stability and cell survival in response to replication poisons. Insight into the importance of PARP during normal cellular replication will continue to be a crucial aspect of research to better understand the family’s diverse role in cellular homeostasis [54,55].

## 7. PARP Family Inhibition in Cancer Therapeutics

The prevention of genomic instability is necessary to restrain oncogenic transformation, cancer development and tumor progression [50]. The importance of DNA damage repair responses in the maintenance of genome integrity is made clear by the vast number of disabled repair factors present across various cancers. Although the deregulation of the DDR appears advantageous to cancer progression, its threat to faithful DNA replication can be exploited for therapeutic value [56]. Through PARP’s expansive role across genomic stability and DNA damage response pathways, its inhibition has become an increasingly relevant therapeutic target. PARP inhibitors have the potential to work as both mono and combinatorial agents, creating synthetic lethality in HR repair-deficient tumors and sensitizing cells to chemotherapeutics or replication stress inducers [56].

PARP inhibitors (PARPi) are a class of orally administered anticancer drugs that compete against NAD+ for PARP’s catalytically active site [9]. PARP1 is the main target of these inhibitors, but the shared homology of the active site in PARP2, PARP3 and PARP4 renders them targets of inhibition as well [18]. Due to this homology, designing selective PARP inhibitors has posed a challenge for researchers, and development is still in progress. The characterization of MART inhibition is limited, predominantly due to their small molecular size, but the exploration of their functions alongside the development of selective inhibitors is well underway [21].

Currently, there are over 250 clinical trials utilizing PARP inhibition, while PARP inhibitors—Niraparib, Rucaparib, Talazoparib and Olaparib—have been FDA-approved [57]. These inhibitors primarily work to strengthen the treatment of breast, ovarian, pancreatic and prostate cancers possessing platinum resistance and/or HR-deficient mutations (predominantly BRCA) [57,58]. Although all function to inhibit PARP, they exhibit different efficacies in areas such as PARP trapping and allosteric activity, leading to variance in off-secondary targets (Figure 2) [58,59,60,61]. Further elucidation is required to determine if, and how, these effects translate to drug-specific cytotoxicity.

## 8. The Action Mechanisms of PARP Inhibition in the DDR

PARPi mechanisms of action have not yet been fully elucidated. However, the loss of PARP’s enzymatic activity has frequently been associated with an inability to modify chromatin and self-dissociate from DNA through auto-PARylation [63]. This mechanism has become known as PARP trapping and renders repair proteins incapable of binding while preventing replication from proceeding, as pictured in Figure 2 [62]. Inadequate gap repair, collapsed replication forks, irreparable S phase-specific DSBs (due to HR deficiency) and persistent SSBs (due to inefficient BER pathway repair) are all possible avenues of PARPi lethality resulting from the trapped PARP1 enzyme (Figure 2) [9].

PARP1’s role in the BER pathway through the identification of AP sites and the recruitment of essential proteins is well established as its main role in the resolution of SSBs [26]. In the absence of PARP1, cells are incapable of repairing SSBs, and an accumulation of DSBs arises. To repair these breaks and maintain genomic stability, cells rely heavily on the DSB pathway, HR. In cancers containing insufficient HR, resulting from mutations in necessary pathway proteins such as BRCA1 and BRCA2, cells treated with PARP inhibitors are forced to rely on NHEJ to repair DSBs. With PARP suppressed, its role in NHEJ inhibition is lost, allowing for the utilization of this error-prone pathway in which cells become more susceptible to mutations over replication periods, leading to greater genomic instability [63].

Although BRCA mutations and HR deficiency leading to irreparable DSBs have been the primary focus of PARP inhibition, the recent uncovering of its mechanistic involvement in the ligation of Okazaki fragments and replication fork progression has led to hypotheses surrounding additional avenues of lethality.

PARP1 has recently been suggested to sense Okazaki fragments [54,55]. In HR-deficient cells, a lack of Okazaki ligation is a leading cause of genomic instability and cell death due to discontinuous DNA synthesis [51]. Similarly, PARP1 has been implicated in the control of replication speed, and when inhibited, cells show a 1.4-fold increase in pace [64]. High replication speeds are correlated with increased genomic instability derived from an accumulation of ssDNA gaps, proposed to be a product of PARPi, preventing the sensing of unligated Okazaki fragments [65]. The induction of replication gaps caused by improper Okazaki fragment processing and repair has become a significant point of interest in PARP inhibition, especially alongside the inhibition of other vital proteins, resulting in synthetic lethality.

## 9. PARP Inhibition Contributes to Synthetic Lethality

Synthetic lethality occurs when two interacting genes, which allow for cell viability when individually suppressed, cause cell death when simultaneously perturbed [66]. The ability to exercise this lethality as a clinical cancer therapeutic has widespread potential. The targeting and inhibition of specific proteins dependent on cell survival in the face of common genetic mutations provide a basis for the utilization of synthetic lethality [66]. As mentioned above, the importance of PARP1 in SSB repair makes PARP1-deficient cells rely on DSB repair (particularly HR) for survival [67]. The enzymatic nature of PARP1, along with its significance in DDR, makes the investigation of the efficacy of PARP1 and HR proteins in synthetic lethality an intriguing topic.

BRCA1/2 play a critical role in the HR pathway and are highly mutated genes in familial breast and ovarian cancers [67]. In 2005, ground-breaking research revealed that BRCA-deficient cancer cells are extremely sensitive to PARP1 inhibitors [67]. As previously mentioned, the inhibition of PARP1 causes SSBs induced by endo- or exogenous sources to result in DSBs, requiring a functional HR pathway to be repaired. Therefore, the absence of key HR proteins like BRCA1/2 causes genomic instability and cell death [67]. Thus, PARP1 inhibition has become a forefront candidate for its clinical efficacy in instigating fatal genomic instability alongside BRCA1/2-defective tumors [66,68].

Although BRCA1/2 has been the focus for synthetic lethality, the loss of other vital proteins in the DNA repair and replication pathways has been shown to sensitize cancer cells to PARP inhibitors [68]. Specifically, proteins involved in DNA replication such as Replication Protein A (RPA), Flap Endonuclease 1 (FEN1) and proliferating cell nuclear antigen (PCNA) have been observed to contribute to synthetic lethality alongside PARP [65,69]. These findings help shine a light on the diverse proteinaceous relationships contributing to synthetic lethality that may be tested as cancer therapeutics going forward [68,70].

RPA is an ssDNA-binding heterotrimeric protein that is required in DNA replication and DNA damage repair pathways and is responsible for protecting ssDNA overhangs, recruiting repair factors and activating cell cycle checkpoints [71]. Once induced by replication gaps, in response to factors such as improper Okazaki processing or DNA damage, RPA coats the ssDNA and protects stalled replication forks, thus activating the ATR pathway [72]. A surplus of replication gaps results in the depletion of the nuclear RPA pool, leading to replication fork breakage and the progression of previously arrested cell cycles. This has come forth as a source of lethality alongside PARPi due to RPA’s implications in maintaining genomic stability under genotoxic stress [65]. Following this correlation, RPA inhibitors have been shown to lead to synthetic lethality in BRCA-deficient cells, indicating the importance of ssDNA gaps as a cause of sensitivity in cancer cells.

FEN1 has a large role in DNA replication and repair; it is responsible for the cleavage of the 5’ flaps left during Okazaki fragment displacement [69]. Importantly, FEN1 also facilitates HR by removing non-homologous DNA ends [73]. In cancers with deficient BRCA1/2 proteins, FEN1 inhibition has been shown to increase sensitivity to DNA damage leading to cell death [74]. In combination with Olaparib, the inhibition of FEN1 leads to increased susceptibility, indicating the gene’s synthetic lethality in HR-deficient cancer cells [69]. Previously thought to only be the result of FEN1’s role in HR, its requirement in the processing of Okazaki fragments may provide insight into its lethality alongside PARP due to their shared role in lagging strand maturation [51,54,55,74].

In addition to FEN1, other proteins involved in the Okazaki fragment maturation process such as Ligase 1 (LIG1) and XRCC1 have been confirmed to be synthetically lethal partners of PARP1 [75,76,77,78]. LIG1 plays an important role in Okazaki maturation by ligating a nick created by FEN1 and finalizing the completion of the nascent strand [76]. PARP1’s role in lagging strand maturation also involves the recruitment of XRCC1 to the SSB when FEN1 and LIG1 strand processing is insufficient [75]. In the absence of PARP1, XRCC1 recruitment is prevented, and the insufficiency further contributes to persistent SSBs and subsequent lethality [79].

PCNA-PARP has been introduced as another promising area to pursue in synthetically lethal partnerships. PCNA’s main role is the assurance of replication longevity [75]. Normally, PCNA controls replication processivity, but the loss of this function deriving from mutated PCNA has been implicated as lethal alongside PARPi [65]. Along with its main function, PCNA plays multiple other roles in DNA replication and replication stress responses. Ubiquitinated PCNA has been implicated in replication fork protection by preventing MRE11 degradation, mediating Okazaki fragment maturation via FEN1 recruitment, and is noted for involvement in the TLS pathway [65,80]. The loss of any of these roles may result in replication gap lesions, consequently leading to unrepairable DSBs in PARP- and BRCA-deficient cells.

The discovery of synthetic lethality and the continual expansion of proteins that can form these relationships alongside PARP, within or outside of BRCA deficiency, prove exciting for future anticancer therapies. The subset listed above is only a small portion of the partnerships emerging. As we begin to further understand the nuances of replication stress responses and DNA damage repair, the combinatorial diversity of these partnerships and their effect on genomic stability within cancer cells will prove significantly important [65].

## 10. PARP Inhibitor Trials and Development across Cancer Treatments

PARPi therapy has primarily been implicated in breast and ovarian cancers due to a high frequency of defective BRCA1/2 mutations, resulting in an increased likelihood of synthetic lethality through a loss of efficient HR [70]. Defective cancers associated with further deficiencies across the HR pathway in proteins such as RAD51, ATR and FANC also show increased cell death when paired with PARPi due to their role as tumor suppressors [70].

Monotherapy involving Olaparib and Talazoparib has been approved for the treatment of mutated germline BRCA (gBRCA) and HER2-negative locally advanced and metastatic breast cancer (BC) [60]. In a recent phase three trial, Olaparib was shown to significantly extend progression-free survival in comparison to single-agent therapies including capecitabine, eribulin and vinorelbine [60]. The trial also noted that toxicity from Olaparib was minor and capable of being treated following dose interruptions, reductions or supporting treatment [81]. A similar phase three trial involving Talazoparib also showed increased progression-free survival in comparison to the standard therapies [82]. Although only a small number of patients stopped treatment due to toxicity, adverse hematological effects were noted in 37.3% of patients [60]. Recently, a study of Olaparib by OlympiA in early-stage gBRCA HER2-negative patients showed improved progression-free survival, indicating a role for PARPi in treating early-stage breast cancer [83].

Like breast cancer, ovarian cancer frequently involves BRCA genes and DDR deficiencies, characterizing up to 50% of high-grade epithelial ovarian cancer patients [84]. Platinum-based chemotherapy has been the forefront treatment for newly diagnosed cases, with FDA approval of combination therapy with the PARP1 inhibitors Olaparib and Bevacizumab. Both PARP inhibitors, in combination with carboplatin-paclitaxel, showed an increase in progression-free survival, with prominent benefits in BRCA-deficient patients [12,84] Both PARP1 inhibitors are also available as monotherapies for stage three and stage four high-grade epithelial ovarian cancer. Niraparib has been shown to improve progression-free survival in NOVA, a phase three clinical study on women with platinum-sensitive ovarian cancer relapse [12]. HR-deficient, non-germline BRCA-mutated patients possessing a loss of heterozygosity, telomeric imbalance and/or large-scale state transitions also showed an increase in progression-free survival following treatment. Additionally, maintenance therapies for breast and ovarian cancer patients have also begun utilizing PARPi [84].

PARP1 activity and expression are also elevated in clear cell renal cell carcinoma in comparison to normal kidney epithelial cells [85]. Some renal cell carcinoma patients have been shown to have altered histone methylase and demethylase proteins, which affect their survival [86]. Interestingly, researchers have recently developed a histone-dependent PARP1 inhibitor that suppresses PARP1 from interacting with complexes, as opposed to arresting a transient complex, as in NAD+ inhibitors [87]. The histone-dependent inhibitors also prevent PARP1-mediated transcription more effectively than NAD+ inhibitors. NAD^+^-like PARP inhibitors in high concentrations have demonstrated decreased survival in normal kidney epithelial cells, while histone-dependent PARPi therapy was active only against the cancer cells, showcasing the clinical possibility of this therapeutic following further testing and development [85,87,88].

Therapies that promote progression-free survival in metastatic castration-resistant prostate cancer (mCRPC) have continued to challenge researchers [89]. In patients suffering from mCRPC, the loss of tumor suppressor genes and DDR repair genes is common and often results in a poorer prognosis, but it also creates a weak point capable of being exploited by PARPi [90]. Olaparib has been shown to improve overall survival in mCRPC cases with ineffective HR through synthetically lethal mechanisms [89]. PARPi, in combination with immunotherapy and chemotherapy, is also undergoing clinical trials for mCRPC, and although unclear, the determination of the efficacy of these therapies is well underway. PARP1’s elevated expression and role in the control of androgen receptors and their gene products in prostate cancer amplify the possible benefit from PARPi [88,91]. The use of histone-dependent PARP inhibitors has also been shown to exert greater antitumor efficacy in both castration-resistant and androgen-dependent prostate cancer than NAD-like inhibitors [91]. Additionally, Phosphatase and Tensin Homolog (PTEN) is a tumor suppressor gene that works in the PI3K/AKT pathway and is commonly mutated in prostate cancer, leading to promoted survival and proliferation pathways for the cancer cells [85]. Recently, PTEN/PI3K pathway inhibitors have been used in combination with PARPi in clinical trials on advanced prostate cancer, hoping to start defining the possible clinical benefits of these combination therapeutics [85].

Like most cancers targeted by PARP inhibition, people with BRCA1 or BRCA2 mutations are susceptible to pancreatic cancer (PC) [92]. Several clinical trials are underway, studying PARPi in mono and combinatorial therapy for PC [92]. In BRCA-mutated metastatic PC, a phase two study found that Olaparib increased progression-free survival in comparison to a placebo group [93]. Clinical trials studying the effects of PARPi on pancreatic ductal adenocarcinoma (PDAC) noted a significant difference between PARPi therapies that is hypothesized to be due to the drug’s PARP trapping potency [94]. In a phase one Talazoparib dose escalation trial with PDAC patients containing BRCA or PALB2 mutations, 20% of participants had partial resection [94]. Focusing on combination therapy, a phase one dose-escalation Olaparib trial including patients with PDAC in combination with irinotecan and cisplatin was stopped due to 89% of patients experiencing grade three or four toxicity [95]. Although the trial was stopped, several patients had positive long-lasting responses, indicating the importance of continuing PARPi trials in PDAC patients. In contrast, a phase one trial conducted on locally advanced pancreatic cancer (LAPC) showed increased overall survival and tolerability in patients receiving combination therapy of gemcitabine, radiotherapy and veliparib [96]. The verification of these results in a future phase two study will be necessary.

The gastric cancer benefit from PARPi has recently been brought to light in both mono and combinatorial therapies, partly due to the extensive amount of DDR and HR deficiencies seen across the cancer [63]. PARPis have been shown to be responsible for a reduction in angiogenesis by lowering the actions of pro-angiogenic factors such as vascular endothelial growth factor (VEGF) [97]. Multiple phase two clinical trials utilizing Olaparib, Talazoparib and niraparib, among others, are underway [63]. Combinatorial therapeutics focused on the loss of effective damage repair by targeting c-MET, Chk1 and PI3K together with PARPi have been shown to have anti-cancer effects. Clinically, Olaparib has been FDA-validated in combination with AZD6788, an ATR inhibitor, and is undergoing clinical trials [63]. Additionally, Olaparib is being studied in combination with an FDA-approved VEGF receptor 2 (VEGFR2) inhibitor, ramucirumab, in advanced gastric cancer.

Non-small cell lung cancer (NSCLC) and PARP inhibition are also being explored clinically [98]. Veliparib, in combination with carboplatin and CDK inhibitors, is currently in phase two and one trials, respectively. Iniparib, although a less potent PARP inhibitor, has had success, albeit not significantly, in increasing the overall survival and progression-free survival of NSCLC patients alongside cisplatin and gemcitabine and is currently in a phase three trial focused on advanced squamous lung cancer [98]. The inclusion of PARP inhibition across a range of clinical cancer trials indicates how impactful the therapeutic benefit of this treatment, alone or in combination, may be across various cancers. As clinical trials continue to begin, and end, it will be important to note each cancer or the mutation’s response to PARPi to direct future anticancer targets and therapeutic regimens.

## 11. PARP1 Sensitivity Biomarkers in Deficient DDR Cells

It is important to note that varying cancers display and respond to the expression levels of PARP1 in the cell differently. Therefore, biomarkers and pre-screening examinations of the patient are important to understanding the benefit or consequences of PARPi therapy [9].

## 12. Some Predictive Biomarkers for PARP Inhibition

The most common biomarker used to predict PARP inhibition sensitivity is mutated BRCA1/BRCA2, resulting in non-functional proteins and deficient HR. As previously mentioned, sufficient HR is required for the maintenance of genomic stability when faced with DSBs [9]. Due to PARP1, PARP2 and PARP3’s role in determining repair response pathways, their inhibition can potentiate the effects of defective HR and fork stalling.

Pictured in Figure 3 are other crucial proteins involved in the DDR, such as RAD51, MRE11, REV7 and EZH2, which also serve as biomarkers of PARPi therapy when mutations cause a loss of function or functional deficiencies increase cell susceptibility to further perturbations [99].

RAD51, a key member of HR, has a paralog, RAD51C, which can be used as a biomarker for PARPi sensitivity [100]. RAD51C is an essential protein necessary for preventing genomic instability through its role in branch migration at sites of DNA damage. Previous studies have found that RAD51C germline mutations are connected to cancer by preventing HR-mediated repair. An increased level of genomic instability and apoptosis was found in RAD51C-deficient cancer cells treated with Olaparib [100]. The increased genomic instability may occur through the reliance on the error-prone NHEJ pathway for the maintenance of RAD51C deficiency-related chromosomal abnormalities [101].

MRE11, an NHEJ and HR protein, plays an important role in the replication stress response [79]. MRE11 is critical for the restart of stalled replication forks due to its role in the MRN complex, which functions to detect and repair DSBs [102]. A loss of functioning MRN complexes, frequently through insufficient MRE11, is commonly found throughout endometrial cancers, proving its capacity as a biomarker for PARPi therapy. HR impairment through non-functional MRE11 has also been shown to sensitize breast, colorectal and hematological cancers to PARPi [102,103,104].

Generally, HR deficiency (HRD) can be scored and used as a PARP biomarker [9]. The score is calculated as the sum of the loss of heterozygosity (LOH), telomeric-allelic imbalance (TAI) and large-scale state transitions (LST). Given that PARP can provide synthetic lethality alongside HR-deficient tumors, the deficiency score may prove to be a reliable indicator of cell sensitivity [9].

In recent years, replication fork protection and stability defects have come to light as an important factor in PARPi lethality [54]. Many HR proteins play a distinct role in replication stress responses and fork protection: BRCA1/2 and RAD51 protect replication forks from nascent DNA over degradation by MRE11. Studies have shown an increased level of degradation in BRCA-deficient cells undergoing PARP inhibition due to fork collapse and subsequent DSBs irreparable by HR [54,62]. In addition, the overexpression of some oncogenes that have roles in replication stress has been shown to be a promising biomarker for PARPi sensitivity.

## 13. Cancer-Specific Predictive Biomarkers for PARP Sensitivity

The MYCN oncogene, present in neuroblastoma, induces replication stress by slowing the replication speed and amplifying fork stalling [105]. When MYCN is highly expressed alongside PARPis such as Olaparib, the replication stress is amplified, resulting in increased cell death. These findings point to MYCN expression as a biomarker for predicting PARP inhibition sensitivity in neuroblastoma patients [105].

The PARP inhibitor Olaparib is FDA-approved to treat gastric cancer; however, phase three trials failed to show a significant improvement in the overall survival of patients [57]. A recent study conducted by a team of researchers at Beijing Proteome Research Center found that metastasis-associated protein 2 (MTA2), which associates with replication origins and compounds the replication stress induced by Olaparib, was found in high levels among gastric cancer tumors [57]. This finding indicates that gastric cancer patients may benefit from MTA2 pre-screening, as high MTA2 may sensitize cells to PARPi therapy by Olaparib.

Deadbox helicase 11 (DDX11) is a biomarker of aggressive renal cell carcinoma, with no expression in healthy kidneys and increasing expression as the stage of renal cell carcinoma increases [106]. Once knocked down, DDX11-deficient cancer cells undergo apoptosis and inhibit proliferation. Importantly, DDX11 knockdown also significantly induced sensitivity to PARPi by Olaparib compared to Olaparib alone, pointing towards DDX11 as a biomarker for PARP therapy in renal cell carcinoma [106].

Prostate cancers containing the overexpression of BCL2 have been noted as being a biomarker for PARP sensitivity as well [107]. BCL2 blocks DSB repair by isolating KU80 in the cytoplasm, which results in cells relying on PARP1-dependent end joining. The reliance on PARP1 indicates the benefit BCL2-expressing prostate cancer patients may experience through PARPi therapies. Additionally, PARPi has been shown to sensitize BCL-overexpressing cancer cells to radiotherapy [107]. 

The use of biomarkers for PARPi across various cancers and DDR repair insufficiencies is beneficial to determining the best course of treatment, but further elucidation is required to develop a full breadth of biomarkers capable of predicting therapeutic sensitivity. Genomic examinations across patients and cancer types should continue to be investigated.

## 14. Variability and Possible Therapeutic Exploitation of Other PARP Family Members

The ability to determine the role and expression levels of various PARP members in different cancer types allows for more direct exploitation when utilizing combination therapeutics and determining which inhibitor may prove most beneficial. Although less understood than other members of the PARP family, PARP4 is thought to be involved in the DDR due to its BRCT domain and can be promiscuously targeted by PARPi therapy due to the shared homology of its regulatory subdomains to those of PARP1 [108,109]. Recent studies have shown that germline mutations in PARP4 may increase the susceptibility of thyroid and breast cancer. Along this line, low PARP4 expression was correlated with a poorer prognosis in a 2016 study conducted by Yuji Ikeda and their colleagues [108].

The tankyrases, distinct from the other PARP family members due to their ankyrin domains, play an impressive role in the Wnt/beta-catenin pathway, which has been proven to be a promising anticancer target [110]. Tankyrases are responsible for targeting the main effector and tumor suppressor of the pathway, AXIN, for degradation. Several tankyrase inhibitors including STP1002 are currently in clinical trials and have shown efficacy in stabilizing AXIN in adenomatous polyposis coli mutated colorectal cancer, leading to a reduction in Wnt target genes [110,111].

PARP6 has also recently been characterized by varying expression levels in different cancers, which is hypothesized to be due to a difference in role depending on the tissue type [9]. PARP6 has notably been correlated with regulating mitosis, and in its absence, centrosomal defects and subsequent apoptosis have been recorded [21]. Following this discovery, Wang et al. utilized a novel PARP6 inhibitor, AZO108, responsible for inhibiting centrosomal clusters, which led to multi-spindle formation [112]. The use of this inhibitor uncovered PARP6 as having a direct role in the modification of Chk1 and mitotic signaling, allowing for in vivo antitumor efficacy in breast cancer cells to be observed [21]. AZO108 may highlight the importance of understanding and targeting other PARP members in anticancer treatments, but the type of cancer being treated, and its relationship with PARP6, must first be understood.

Although PARP7 may be poorly understood, its role in ovarian cancer as a MARylator of alpha-tubulin, leading to microtubule instability and cancer cell motility, indicates the possibility of its inhibition as a new avenue of PARP therapeutics [113]. PARP7 has also been shown to negatively regulate the sensing of nucleic acids in tumor cells, reducing the ability of the immune system to target the tumors [114]. A newly developed selective PARP7 inhibitor, RBN-2397, has been shown to restore the type 1 interferon response, leading to a restoration of anti-tumor immunity. Notably, RBN-2397 has led to tumor regression in murine models and is currently in phase one clinical trials for patients with solid tumors [114,115]. However, PARP7 expression varies depending on the tissue; another study found that its mRNA levels decreased in cancer cells compared to normal cells, with high expression being an indicator of good outcomes in breast cancer [22]. Interestingly, in estrogen receptor (ER)-positive breast cancer, PARP7 has been found to negatively regulate the oncogenic capability of ER alpha, leading to tumor suppression.

Despite PARP10’s role in overcoming replication stress, it has also been shown to quell tumor metastasis through the regulation of cell migration [116]. It does this by MARylating and thus suppressing Aurora A, an often-overexpressed protein thought to impact survival signaling pathways in tumor cells. Interestingly, PARP10 overexpression is commonly seen in cancers due to its role in the alleviation of replication stress, with the appearance of longer replication tracts noted under both control and Hydroxyurea-treated cells [24]. Recent studies detailing the possibilities of PARP10 inhibitors as anti-cancer agents have proven promising, showing that siRNA-targeted PARP10 repressed growth and metastatic capabilities in oral squamous cell carcinoma by negating its ability to regulate cell proliferation and apoptosis [117]. Following this discovery, a potent PARP10 inhibitor, A82-(CONHMe)-B354, has also been developed, although further research is needed to determine its efficacy against tumor cells and its role as a future therapeutic agent [118].

PARP11’s role in stabilizing the nuclear envelope during localization is well known, but it has also recently been linked to the positive regulation of immunosuppressive tumor microenvironments [119,120]. In light of this, a selective PARP11 inhibitor, IKT7, has been developed and will need to undergo further testing and clinical trials to determine its role in anticancer care [121].

PARP14’s role in suppressing replication stress via HR-mediated repair makes it an intriguing topic of discussion for inhibition [52]. RBN012759, a newly developed selective PARP14 inhibitor focused on PARP14’s role in immunotherapy, has shown a reversion of IL-4-directed pro-tumor genes, indicating the potential to employ the inhibitor as an anticancer therapeutic. Although more research is needed, it will be fascinating to observe the effect of PARP14 inhibition on the DNA damage response and replication stress both as a monotherapy or as a participant in synthetic lethality [52,122].

Interestingly, PARP16 has been shown to be a target of the PARP1 inhibitor Talazoparib when used as a therapeutic for small cell lung cancer, showing the range of Talazoparib and depicting a larger view of its mechanism of action [123]. In ovarian cancer, an upregulated cytosolic NAD+ synthase leads to increased activity of PARP16, resulting in translation and other cellular processes [124]. Upon the depletion of PARP16, ovarian cells showed reduced proliferation and increased protein-specific translation, revealing how PARP16 can potentiate cancer cell homeostasis through MARylation, resulting in a honing of protein synthesis [124].

## 15. PARP Inhibitors Resistance

Although the role of the PARP family and the therapeutic value of its inhibition in cancer are continuously being uncovered, the clinical promise has been dampened by the occurrence of drug resistance [59,125]. The most documented mechanism of resistance is the restoration of HR through BRCA reversion mutations that restore wildtype or hypomorphic BRCA1 or BRCA2 functions in the cell, allowing for proficient HR [59,125]. In the absence of defective HR, the synthetic lethality between BRCA1/2 mutations and PARP inhibition previously inducing cell death is lost, alongside clinical drug effectiveness [126]. Reversion mutations allowing for the induction of adequate BRCA1/2 reading frames are hypothesized to result from the increased genomic instability leading to base substitutions, insertions or deletions [125,127]. These alterations then ‘revert’ BRCA’s open reading frame to a functional or semi-functional sequence [126,127].

Interestingly, BRCA-independent restorations of HR including the repression of the NHEJ pathway have also been shown to result in PARPi resistance [128]. 53BP1 is a protein involved in inhibiting the nucleolytic end resection required for HR and thus promotes NHEJ for repair [38,127]. A loss of or lowered expression of 53BP1 have been found in triple negative and BRCA-deficient breast cancer. 53BP1 deficiency causes a shift between DDR pathways that allows for the promotion of HR through BRCA1-independent end resection, thus providing resistance to PARP inhibition [128].

The upregulation of RAD51 and its subsequent foci formation have also been documented in PARPi resistance, indicating its ability to compensate for the loss of BRCA in HR [129,130]. On a similar note, cancers with the deficient HR genes RAD51C, RAD51D and PALB2 being treated with PARP inhibitors subjected to secondary mutations allowed for the induction of proper, or semi-proper, protein functioning, leading to a restoration of sufficient HR and PARPi resistance [130,131,132].

The restoration of replication fork protection has also been shown to confer resistance to PARPi therapeutics by improving genomic stability [133,134]. It is well established that BRCA1 and BRCA2 play a significant role in the protection of stalled forks, preventing the nucleolytic degradation of the nascent DNA by MRE11 and MUS81, among others [135]. In the presence of defective BRCA1 and BRCA2, replication fork protection is decreased, and DNA is more susceptible to degradation [134]. EZH2 is a key enzyme that mediates the recruitment of MUS81 to replication forks and has been implicated in the promotion of fork breakdown. When the expression of EZH2 is low, MUS81 is not recruited, and fork stabilization is improved, causing clinical chemotherapy and PARPi resistance in BRCA2-deficient cells [132,134]. In BRCA2-mutated cells, a similar protein called PTIP, which is also responsible for nuclease recruitment to replication forks, is downregulated, causing PARPi resistance in BRCA-1-deficient cells through the protection of nascent DNA [134,136].

Another proposed mechanism of PARPi resistance is the restoration of PARP1 signaling through the loss of PAR glycohydrolase (PARG), which is responsible for the degradation of PAR chains [137]. It has been hypothesized that the downregulation of PARG removes a layer of reinforcement that acts to prevent PAR formation and trapping, indicating that PARPi’s effect is partly due to the help of PARG working in a similar manner [137]. The depletion of PARG has been shown to lower the amount of PARP1 trapping on DNA, thus counteracting PARPi therapy in an HR-independent fashion [132,137].

Various other mechanisms of resistance such as point mutations in PARP1 leading to reduced PARP trapping, the upregulation of drug efflux pumps such as Abcb1a/b and the dysregulation of signaling pathways such as PI3K/AKT have been documented as well [132,138,139].

Resistance to PARPi is a growing challenge across many forms of cancer and highlights the many ways in which we are constantly at tug of war with the disease. Although resistance is a concern and may lead to a loss of therapeutic effectiveness, it may also give insight into the development of novel combinatorial therapeutics as well as aid in understanding the most effective ways to incorporate PARPi as an anticancer agent.

## 16. Overcoming PARPi Resistance

To overcome PARPi resistance, the mediation of synthetic lethality through mechanisms such as immune checkpoint inhibitors, cancer-dependent target therapies, cell combination and DDR inhibitors is becoming increasingly important [59].

The role of immunotherapies in cancer treatments is being vastly investigated. Particularly, the advancements in immune checkpoint inhibitors, chimeric antigen receptors and TCR-engineered T cells have been showing clinical promise [140,141]. Interestingly, HR-deficient cancers have been documented as harboring an increased number of tumor-specific neoantigens, leading to an increase in the immune response [59,142]. Anti-PD-1 antibodies, which target the inhibitory ligand PD-L1 on tumor cells and thus increase the cell functioning previously burdened by PD-L1, are FDA-approved and have been shown to potentiate the effects of PARPi through crosstalk [141,142]. This is backed by the discovery that PARPi upregulates PD-L1 through GSK3B inactivation, providing a basis for combined therapies of PARPi and PD-1. Clinical trials exploring the interplay between PD-1 antibodies and PARPis are underway for extensive-stage small-cell lung cancer, recurrent and metastatic endometrial cancer and ovarian cancer [143,144,145].

PARPi has also been implicated in combination with PI3K inhibitors due to the role of PI3K in oncogenic signaling, leading to cancer cell proliferation and survival [138,146]. PARPi therapy is shown to increase the activation of the PI3K pathway, suggesting that the pathway may attenuate the PARPi efficacy and cause resistance [138,147]. The combination of PI3K and PARP inhibitors has shown promise in Talazoparib-resistant triple-negative breast cancer by suppressing proliferation and inciting apoptosis [148]. A second PI3K/histone deacetylase (HDAC) inhibitor synergized with Olaparib in small cell lung cancer provides more of a basis for clinical trials [149].

The topic of HDAC inhibitors in the fight against PARPi resistance is part of a large area of focus surrounding the combination of PARPis with other DDR proteins with roles in HR [150]. HDAC inhibitors have been shown to sensitize cells to PARPi in triple-negative breast cancer due to an increased dependence on the error-prone NHEJ repair pathway [151].

The regulation of the cell cycle by ATR and CHK1 kinases is crucial for managing replication stress, working to arrest the cell cycle and contributing to fork protection and proficient HR repair [152]. The importance of ATR and CHK1 in areas of the DDR that PARPi targets makes it a promising candidate, and current clinical studies focusing on the inhibition of these kinases in replication fork-stabilized and HR-sufficient PARPi-resistant cancers are underway [152,153,154].

## 17. Future Directions

Thus far, the majority of clinical PARP inhibitors have targeted PARP1, which affects PARP1’s role in the detection and resolution of DNA damage [9,18]. Due to a shared homology, PARP2, PARP3 and PARP4 are often also affected by these agents. As new and important roles of other PARP family enzymes come to light, it will be important to develop potent and selective inhibitors capable of targeting other, individual PARP enzymes. As previously mentioned, the prospect of selectively targeting PARP enzymes has widespread potential such as blocking PARP10’s role in the alleviation of replication stress and preventing PARP7 from downregulating the immune response to tumors [114,116]. Given the anticancer effects inhibition these PARP family enzymes could have, ensuing research will be important to creating and bringing these inhibitors to the clinic.

Similarly, the available biomarkers for PARPi account for only a fraction of the possible mutations and interactions that gauge for therapeutic sensitivity. Further biomarker testing across cancer types and within specific patients should be researched to better highlight the targets and cellular responses of PARPi. Biomarkers for specific PARP family members should also be addressed to bring the scope of potential PARPis throughout various cancers to light. As research on the development of selective inhibitors continues, it will be crucial to have data pertaining to cancer signatures that may show the best clinical response.

As resistance mechanisms opposing PARPis’ therapeutic potential continue to emerge, the optimization of combinatorial PARPi therapy as a counter mechanism is needed now more than ever. Recently, PARPi has been shown to sensitize tumor cells to immune checkpoint inhibitors through the modulation of the tumor microenvironment in ovarian cancers, hinting at the therapeutic value this combination could have [155]. Similarly, a new phase one clinical trial is underway, investigating PARPi Talazoparib alongside the DNA Methyltransferase inhibitor Decitabine in relapsed Acute Myeloid Leukemia, which was reportedly well tolerated by patients [156]. Advancements such as these will continue to be a tug of war against drug resistance, but each one leads us closer to clinically beneficial therapies for patients everywhere.

## 18. Concluding Remarks

Through years of research, the role of the PARP family as a homeostatic regulator and genome protector has been extensively examined. The enzyme family’s diverse range of functions includes regulating replication, detecting and fixing DNA damage and responding to replication stress [9]. Due to this vast range, and the interplay between PARP and genomic stability, its inhibition has been implicated as an anticancer therapeutic. Primarily, PARPi has contributed to a synthetically lethal partnership in BRCA1/2-deficient cancers, although it has been shown to partner with many other proteins in a similarly lethal fashion [70]. Today, there are over 250 PARPi clinical trials underway, with many showing promising results in progression-free survival across a variety of cancer types [57,63,84,87,89,98]. Despite the promise PARPi has shown as a clinical anticancer agent, it has been met with resistance in a number of mechanisms from the restoration of HR to point mutations in PARP itself, leading to pitfalls in the current PARPi market [59,125,132]. These challenges have yet to defer researchers, and methods of countering therapeutic resistance are well underway. Showing our defiance, several counter-resistant treatments such as combining PARPi with immunotherapy and other DDR inhibitors have been introduced with encouraging results [59]. The study of PARPi as an anticancer agent is clinically significant now more than ever before due to the optimistically treatable, global health concern cancer presents. The development of more efficient therapies with the maintenance of therapeutic windows is vitally important to quelling the increasing burden that cancer domineers across the globe.

## Figures and Tables

**Figure 2 cells-12-01904-f002:**
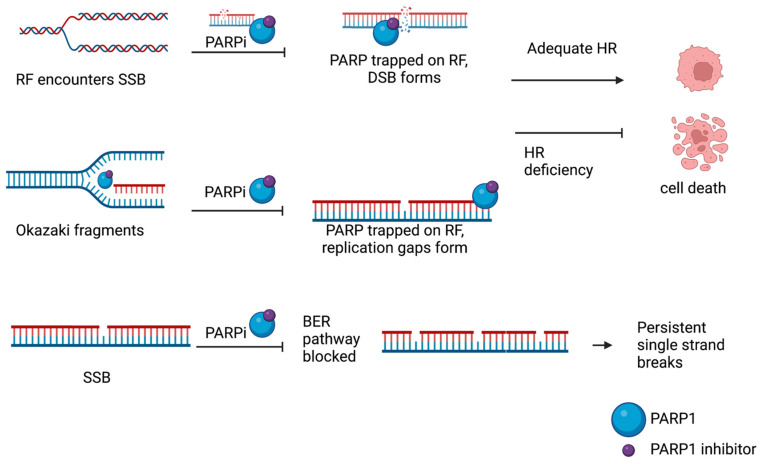
Key mechanisms proposed for PARP inhibition in replication include PARP trapping, resulting in DSBs [62], an inability to ligate Okazaki fragments, leading to replication gaps [51,55], and an inability to utilize the BER pathway, leading to persistent SSBs [9].

**Figure 3 cells-12-01904-f003:**
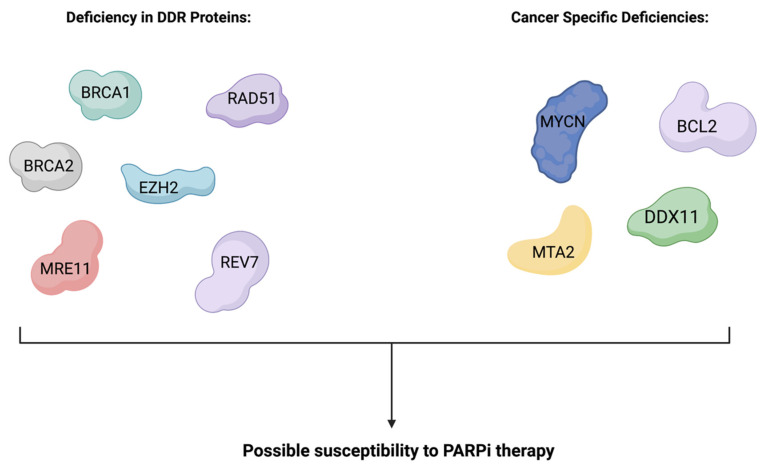
Some PARP sensitivity biomarkers applicable to DDR-deficient and cancer-specific cell phenotypes. These include deficiencies in BRCA1, BRCA2, RAD51, EZH2, MRE11, REV7, MYCN, BCL2, DDX11 and MTA2 [9,79,100]. This figure is not all-encompassing, as new markers are continually being elucidated in connection with PARP-sensitized cancer cells.

**Table 1 cells-12-01904-t001:** The enzymatic activity and subcellular locations of PARP family members [1,2,3,4,5,6,7,8,11,12,13,14].

Enzyme Name	ADP Ribosylation	Cellular Location	Key Functions
PARP1	PAR	Nucleus	DNA damage response (detection, regulation, recruitment) + chromatin remodelling
PARP2	PAR	Nucleus, cytoplasm	DNA damage response + chromatin remodelling
PARP3	MAR	Nucleus, cytoplasm	DNA damage response + chromatin remodelling
PARP4	MAR	Cytoplasm and nucleus	Protein regulation, cellular transport
PARP5a	PAR	Cytoplasm	Telomere length, vesicle trafficking
PARP5b	PAR	Cytoplasm	telomere length, vesicle trafficking
PARP6	MAR	Cytoplasm	regulation of MPS induction, cellular adhesion and motility
PARP7	MAR	Nucleus, cytoplasm	regulation of gene transcription, stress responses, innate immunity
PARP8	MAR	Nuclear envelope, cytoplasm	not yet established
PARP9	INACTIVE	Nucleus	not well understood, but thought to be DNA repair
PARP10	MAR	Cytoplasm, lesser extent nucleus	
PARP11	MAR	Nuclear pores	nuclear envelope stability/remodelling/spermatid formation
PARP12	MAR	Golgi, cytoplasm	golgi maitenance/cellular stress response
PARP13	INACTIVE	Cytoplasm	promotes degradation of viral mRNA
PARP14	MAR	Cytoplasm, nucleus	regulates cyotskeletal structure, responds to replication stress/ DNA damage, inflammatory signalling pathways
PARP15	MAR	Unknown	unknown
PARP16	MAR	Cytoplasm, endoplasmic reticulum	mitotic regulation/ER stress sensor regulation

## Data Availability

Not applicable.

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
