# Peer review of "Unravelling the Role of PARP1 in Homeostasis and Tumorigenesis: Implications for Anti-Cancer Therapies and Overcoming Resistance"

_cells, 2023, doi:10.3390/cells12141904_

Round 1

Reviewer 1 Report

The review article entitled “Life on the front lines: An overview of the PARP family enzymes evolving role in genomic stability and cancer therapeutics” is a well written review outlining the role that PARPs (specifically PARPs 1-5) and PARP inhibitors play in cancer therapeutics. I would recommend publishing this article with the following minor modifications:

1)    Think carefully about changing the title of this article. From reading the current title, it sounds like the review will focus on all members of the PARP family and how they relate to cancer. In reality, the review focuses 95% of the time on PARP1 and it’s inhibitors with very little actually dedicated to other members of the PARP family.

2)    On page 2 the authors describe the catalytic role for many of the PARP family of enzymes, but I believe that both PARP13 and PARP9 lack catalytic activity.

3)    On page 2 the authors also mention the lack of specific inhibitors of MARTs when in reality there are actually several. Good review on this topic - Journal of Medicinal Chemistry  20226511, 7532-7560.

4)    The figure on page 4 is a bit hard to read, font too small. Please adjust accordingly.

5)    On page 13, the authors mention that the homology between PARP1 and PARP4 is strong and therefore many PARP 1 inhibitors also inhibit PARP4, but I think this is a poor generalization. Please see the following manuscript that shows this data for all of the clinical PARPi’s. J. Med. Chem. 2017, 60, 1262−1271.

The one section that actually discusses other members of the PARP family should be expanded. For example, there are several papers on PARP10 inhibitors and PARP14 inhibitors and their potential role in cancer.

Author Response

First off, we wanted to note our gratitude to all of the referees for their insights, scrutiny, and constructive feedback on our manuscript. We feel as though the quality of our work has been enriched and the story is more comprehensive. Please see below for our responses to the referees’ comments.

Reviewer #1

  1. Think carefully about changing the title of this article. From reading the current title, it sounds like the review will focus on all members of the PARP family and how they relate to cancer. In reality, the review focuses 95% of the time on PARP1 and its inhibitors with very little actually dedicated to other members of the PARP family.

Thank you for your comment. We have now changed the title to “Unravelling the Role of PARP1 in Homeostasis and Tumorigenesis: Implications for Anti-Cancer Therapies and Overcoming Resistance”.

  1. On page 2 the authors describe the catalytic role for many of the PARP family of enzymes, but I believe that both PARP13 and PARP9 lack catalytic activity.

Refer to Table 1 and Table 1 legend for corrected information regarding PARP9’s catalytic activity. Please page 2.

  1. On page 2 the authors also mention the lack of specific inhibitors of MARTs when in reality there are actually several. Good review on this topic - Journal of Medicinal Chemistry  2022, 65, 11, 7532-7560.

Thank you for this comment, since initially writing this review additional papers touching on the creation and availability of MART inhibitors have been published. Therefore, to the best of our knowledge, we have updated our current work to represent the newest information regarding these inhibitors. To specifically address this comment, we have made highlighted changes on page 3 as well as reiterating these points.

  1. The figure on page 4 is a bit hard to read, font too small. Please adjust accordingly.

The font size has been adjusted accordingly. Refer to Figure 1 on page 4.

  1. On page 13, the authors mention that the homology between PARP1 and PARP4 is strong and therefore many PARP 1 inhibitors also inhibit PARP4, but I think this is a poor generalization. Please see the following manuscript that shows this data for all of the clinical PARPi’s. J. Med. Chem. 2017, 60, 1262−1271.

Thank you for pointing out the discrepancy between the current literature and our proposed work. Page 13 and page 14 have been updated to explain promiscuity more carefully in PARP inhibitors and why PARP4 can be targeted by PARP1 inhibitors.

  1. The one section that actually discusses other members of the PARP family should be expanded. For example, there are several papers on PARP10 inhibitors and PARP14 inhibitors and their potential role in cancer.

We would like to thank the reviewer for this comment and have addressed the paper accordingly. Portions of information regarding PARP10 have been added on Page 14 while information regarding PARP14 has been added on Page 6. Information already contained within the paper regarding inhibitors of these PARP family members and their newly highlighted role in various cancers has been reconfirmed to align with the updated literature.

Reviewer 2 Report

Lovsund et al., review article is well wrote and very clear about the take-home message. PARP family enzymes are many and for some of them not many information are available in literature. Instead, PARP1 is very well characterized among all. For this reason, I think some information about other DNA Damage Response pathways are missing.

Major comments:

During the first chapters pag 4 and Figure 1, authors are describing a role for PARPs after UV-damage during BER and DSBs repair pathways, also NER pathway has an essential participation during UV-induced damage.

Therefore, I would add a chapter about it, I can put here some references (Chauduri et al., 2017; Pines et al., 2012; Yi et al., 2019; Fischer et al., 2014;...)

Talking about PARPs (in particular PARP1), DSBs and BRCA1 (chapter 4 pag 5) I would add some references missing like a direct effect of parylation on BRCA1 (Hu et al., 2014; Cong et al., 2021; Lodovichi et al., 2023;...) 

In the middle of page 17, authors need to add all the paragraph informations about: author contributions, Funding, data availability statement and conflicts of interest

Author Response

During the first chapters pag 4 and Figure 1, authors are describing a role for PARPs after UV-damage during BER and DSBs repair pathways, also NER pathway has an essential participation during UV-induced damage.

Therefore, I would add a chapter about it, I can put here some references (Chauduri et al., 2017; Pines et al., 2012; Yi et al., 2019; Fischer et al., 2014;...)

Talking about PARPs (in particular PARP1), DSBs and BRCA1 (chapter 4 pag 5) I would add some references missing like a direct effect of parylation on BRCA1 (Hu et al., 2014; Cong et al., 2021; Lodovichi et al., 2023;...) 

To address this comment, as the reviewer suggested, we have expanded our SSB/DNA damage repair section to include the NER pathway as well as explaining PARP’s role within the said pathway. We have also added the aforementioned references on page 5 to ensure adequate references and previous contributions.

  1. In the middle of page 17, authors need to add all the paragraph information about: author contributions, Funding, data availability statement and conflicts of interest

We would like to thank the reviewer for pointing this out. A paragraph of information about author contributions, Funding, data availability statement and conflicts of interest has been now added.
